# Changes in Neutrophil Count During Valganciclovir Therapy for Symptomatic Congenital Cytomegalovirus Infection

**DOI:** 10.3390/biomedicines13071739

**Published:** 2025-07-16

**Authors:** Aoi Kawamura, Shinya Abe, Keisuke Shirai, Yu Masuda, Yukihito Imagawa, Yuki Nakata, Takumi Kido, Mariko Ashina, Hisayuki Matsumoto, Kenji Tanimura, Yasumasa Kakei, Takumi Imai, Kandai Nozu, Kazumichi Fujioka

**Affiliations:** 1Department of Pediatrics, Kobe University Graduate School of Medicine, Kobe 650-0017, Japan; sst888.aoi@gmail.com (A.K.); sky.my.kh@gmail.com (S.A.); ksk1024@med.kobe-u.ac.jp (K.S.); yumasuda1@gmail.com (Y.M.); yukki940205@gmail.com (Y.I.); yuki111@med.kobe-u.ac.jp (Y.N.); taku32619@gmail.com (T.K.); marikoa@med.kobe-u.ac.jp (M.A.); nozu@med.kobe-u.ac.jp (K.N.); fujiokak@med.kobe-u.ac.jp (K.F.); 2Department of Clinical Laboratory, Kobe University Hospital, Kobe 650-0017, Japan; hisa-mt@live.jp; 3Department of Obstetrics and Gynecology, Kobe University Graduate School of Medicine, Kobe 650-0017, Japan; taniken@med.kobe-u.ac.jp; 4Clinical and Translational Research Center, Kobe University Hospital, Kobe 650-0017, Japan; imai@med.kobe-u.ac.jp

**Keywords:** neutrophil nadir, neutropenia, valganciclovir, congenital cytomegalovirus infection

## Abstract

**Background/Objectives**: Neutropenia is a common adverse effect of oral valganciclovir (VGCV) treatment in infants with congenital cytomegalovirus infection (CCMVI), with an estimated prevalence of 20%. However, its clinical course and associated factors, including the influence of VGCV dosage, remain inadequately characterized. **Methods**: We conducted a single-center retrospective cohort study of infants treated with VGCV for symptomatic congenital CMV infection (CCMVI) at the Kobe University Hospital between 1 April 2009 and 31 March 2017. Detailed descriptive analyses of neutropenia were performed, and factors associated with its onset were explored using univariable logistic regression analyses. **Results**: A total of 31 patients were included, and neutropenia occurred in 35% of them during the 6-week treatment period. Its occurrence was observed throughout the treatment course, with no substantial difference in incidence between the 16 mg/kg/day and 32 mg/kg/day dosing groups. Neutropenia was more likely to occur in infants with shorter gestational age. **Conclusions**: Neutropenia occurred in 35% of patients during 6 weeks of VGCV treatment, irrespective of dosage, and was more common in those with shorter gestational age.

## 1. Introduction

Congenital cytomegalovirus infection (CCMVI) is the most common congenital infection caused by transplacental transmission, with a reported incidence of 0.3–0.7% [1,2,3]. Approximately 40–60% of symptomatic cases of CCMVI develop neurodevelopmental sequelae, with sensorineural hearing loss (SNHL) being the most common, followed by cognitive impairment and cerebral palsy [4]. In recent years, valganciclovir (VGCV) treatment for CCMVI in the early postnatal period has been shown to improve neurodevelopmental outcomes [5,6,7,8,9,10]. Previously, our group revealed that VGCV treatment for either 6 weeks or 6 months could improve mild-to-moderate SNHL but was ineffective for severe cases [11]. More recently, we showed that initiation of VGCV treatment within 2 months of age was as effective as initiation of treatment within 1 month, which is the classical treatment time limit [12].

The known side effects of oral VGCV treatment include neutropenia, anemia, and thrombocytopenia due to myelosuppression, as well as liver dysfunction, pancreatitis, and renal dysfunction [5,13,14]. Neutropenia has been reported to occur in approximately 20% of treated cases [5,14], and a substantial amount of these cases require drug cessation owing to the high risk of serious infection. Based on the results of recent analyses from our multi-center prospective studies using a 6-month regimen of VGCV dosed at 32 mg/kg, 62.5% of infants experienced their lowest neutrophil count within the first 6 weeks of treatment [15]. Meanwhile, reducing the VGCV dosage has been suggested as a treatment approach for neutropenia; however, its effectiveness in alleviating neutropenia has yet to be clarified [16].

In contrast, a case report indicated VGCV treatment was successfully completed by administering granulocyte-colony stimulating factor (G-CSF) to account for neutropenia [17], implying that neutropenia can be controlled by G-CSF. Nevertheless, there is no clear correlation between the degree of neutropenia and blood concentration of VGCV [12], and the changes in neutrophil count during VGCV treatment remain to be elucidated.

This study aims to provide a detailed descriptive analysis of neutropenia and to explore factors related to its onset, including the potential association between valganciclovir (VGCV) dosage and neutropenia.

## 2. Materials and Methods

### 2.1. Study Design and Patients

This single-center retrospective cohort study included infants who were treated with VGCV for symptomatic CCMVI at the Kobe University Hospital from 1 April 2009 to 31 March 2017. Infants aged > 60 days at the start of VGCV treatment, those with other congenital anomalies, and those who deviated from the treatment protocol described below were excluded from the study. For each case, patient background, changes in neutrophil count up to 6 weeks consecutively after the start of VGCV therapy, history of suspension/reduction/discontinuation of VGCV treatment, G-CSF use, and CMV DNA titers in the plasma and urine before treatment were collected from medical records. As part of a prospective cohort study for universal congenital cytomegalovirus (CMV) screening at the Kobe University Hospital [18], we examined CMV DNA from urine samples collected on filter paper within the first week of life from all newborns born at our hospital and measured urine viral loads using quantitative reverse transcription polymerase chain reaction (qRT-PCR) in neonates suspected of having CCMVI [19]. To clarify the characteristics of the study population, background information on asymptomatic CCMVI infants will also be presented as external control data. This study was approved by the Ethics Committee of the Kobe University Graduate School of Medicine (approval numbers: 923, 1214), and written informed consent was obtained from the guardians of all participants.

### 2.2. Diagnostic Criteria

CCMVI was diagnosed based on a positive CMV-DNA qRT-PCR result from a liquid urine sample within 21 days of birth. According to our previous reports [19,20,21], symptomatic CCMV was defined as having at least one of the following symptoms: small for gestational age (SGA) (birthweight < 10th percentile for gestational age based on sex-specific Japanese standards [22]); microcephaly (head circumference < −1.5 the standard deviation of the mean value for Japanese newborns of the same gestational age [22]); thrombocytopenia (platelet count < 1 × 105/µL); liver dysfunction (serum aspartate aminotransferase level > 100 U/L); eye complications (CMV-associated retinopathy, such as chorioretinitis, diagnosed by a pediatric ophthalmologist); brain computed tomography or magnetic resonance imaging abnormalities (such as intracranial calcifications, ventricular dilation, white matter abnormalities, and cortical dysplasia) as diagnosed by a radiologist; or unilateral or bilateral hearing dysfunction, diagnosed based on auditory brain stem response abnormalities using a Neuropack S1 (Nihon Kohden Co., Tokyo, Japan) (including absent wave V to 40 dB or 50 dB at a postconceptional age of 37 weeks or 34–36 weeks, respectively) [11,23].

### 2.3. Treatment Protocol

Between April 2009 and December 2015, oral VGCV doses of 32 mg/kg/day for 6 weeks were adopted, and from January 2016, oral VGCV of 32 mg/kg/day for 6 months were adopted as an oral treatment for symptomatic CCMVI in our institute [5]. Among CCMVI types, central nervous system-localized types in which complications were limited to brain imaging abnormalities, hearing dysfunction, and eye complications were treated with a reduced dosage regimen (VGCV 16 mg/kg/day) [24,25]. Complete blood counts and biochemistry were evaluated before treatment and weekly for 6 weeks after treatment. If neutrophil count became <500/mm^3^ during VGCV treatment, treatment was interrupted. Neutropenia onset was defined as a decrease in the neutrophil count falling to <500/mm^3^. Administration was resumed when neutrophil counts recovered to ≥750/mm^3^. Thereafter, if the neutrophil count decreased to <750/mm^3^ again, the dosage was reduced to 50%. If neutrophil count became <500/mm^3^ again, treatment was discontinued. In cases of neutropenia, G-CSF was administered at the discretion of the attending physician.

### 2.4. Statistical Consideration

Given the retrospective study design and the rarity of the target disease, all patients that met the eligibility criteria were included as study participants. Based on a rough prior estimate of the number of neutropenia events, we assumed that it would be feasible to explore factors associated with neutropenia using at least univariable analysis.

Descriptive data were presented as median (range, i.e., minimum to maximum) for continuous variables, and as number (percentage) for binary variables. Time to neutropenia onset, defined from the initiation of treatment, was described using the Kaplan–Meier method. To explore patient factors associated with the onset of neutropenia within 6 weeks after the initiation of treatment, univariable logistic regression analysis was performed using the presence or absence of neutropenia as the outcome. A two-sided *p*-value < 0.05 was considered statistically significant. All statistical analyses were performed using R software, version 4.4.2 (R Foundation for Statistical Computing, Vienna, Austria).

## 3. Results

### 3.1. Patient Characteristics

During the study period, 31 infants were enrolled in this study (Table 1). Among these infants, 20 started VGCV treatment before December 2015, and 11 were administered VGCV 16 mg/kg/day because of central nervous system-localized types, as detected on brain imaging abnormalities, ophthalmological complications, or auditory brain stem response (ABR) abnormalities alone. Eleven infants received VGCV treatment after January 2016. Accordingly, 11 patients received VGCV 16 mg/kg/day for 6 weeks, while 20 received VGCV 32 mg/kg/day for 6 weeks as initial treatment. The mean gestational age was 36 (range, 30–40) weeks and mean birthweight was 2210 (940–3312) g. In addition, we included asymptomatic CCMVI infants who did not require VGCV treatment but were managed during the same period as an untreated control group.

Before treatment, the median plasma CMV DNA level was 5.6 × 102 (2.1 × 10^0^–9.3 × 10^4^) copies/106 white blood cells, and the median urinary CMV DNA level was 5.6 × 10^7^ (1.0 × 10^3^–1.7 × 10^9^) copies/mL. Various clinical symptoms, including microcephaly (*n* = 9, 29%), SGA (*n* = 11, 35%), thrombocytopenia (*n* = 15, 48%), liver dysfunction (*n* = 9, 29%), brain imaging abnormalities (*n* = 28, 90%), CMV-related eye complications (*n* = 7, 23%), and ABR abnormalities (*n* = 22, 71%), were noted. In the untreated control group, fewer complications were observed compared to the treatment group, and there was a tendency toward lower neutrophil counts and viral loads. However, because the timing of sample collection differed between the two groups, a direct comparison could not be made.

### 3.2. Description of Neutropenia Onset During 6 Weeks of VGCV Treatment

During 6 weeks of VGCV treatment, neutropenia onset (neutrophil count fell to <500/mm^3^) was observed in 11 patients (35%) (Figure 1, Table 2). Seven patients (23%) required drug interruption due to neutropenia after starting VGCV treatment. Additionally, nine patients (29%) received G-CSF treatment for neutropenia. Regarding the time to first onset of neutropenia, occurrences were most frequent on the first day of VGCV treatment, but incidences were observed throughout the 6-week period (Figure 1). Even when focusing on the neutrophil nadir in individual patients, the timing of the nadir was distributed across the entire 6-week period (Table 2). The detailed neutrophil count trajectories in individual patients are shown in Figure 2.

### 3.3. Factors Associated with the Onset of Neutropenia Within 6 Weeks After the Initiation of Treatment

The results of the series of univariable logistic regression analyses to explore patient factors associated with the onset of neutropenia are presented in Table 3. A longer gestational age was negatively associated with neutropenia onset (odds ratio per 1-week-increase in gestational age: 0.72, 95% confidence interval: 0.54–0.97, *p* = 0.031). We performed a similar logistic regression analysis with the occurrence of neutropenia after two weeks of VGCV treatment as the outcome, which yielded generally consistent results (Appendix A). As a post hoc analysis, the distribution of gestational age was plotted for patients with and without neutropenia at 6 weeks, as shown in Figure 3. The median gestational ages for patients with and without neutropenia were 35 and 38 weeks, respectively. The *p*-value from the Wilcoxon rank-sum test comparing the two groups was 0.019.

## 4. Discussion

To our knowledge, this study is one of the few reports that have investigated factors associated with neutropenia in children with CCMVI treated with valganciclovir (VGCV). Neutropenia occurred in 35% of patients during the 6-week period of VGCV treatment. Its occurrence was observed throughout the treatment period, and there was no substantial difference in incidence between the 16 mg/kg/day and 32 mg/kg/day dosing groups. In addition, neutropenia was more likely to occur in patients with shorter gestational age.

Neutropenia is the most frequent complication of oral VGCV treatment for CCMVI [5,26]. Previous reports have investigated the frequency of neutropenia in children with CCMVI who received 6 weeks of ganciclovir treatment [27], children with CCMVI who received 6 weeks of VGCV treatment [13], and children with CCMVI who received 6 months of VGCV treatment [5]; however, to date, no study has revealed the changes in neutrophil counts throughout the 6 weeks period after VGCV treatment. Based on the results of recent randomized controlled trials, oral VGCV was administered for 6 months as an antiviral treatment in children with symptomatic CCMVI [26]. According to Kimberlin et al., neutropenia occurred in 19% of infants with CCMVI within the first 6 weeks during their 6 months of VGCV treatment, and 2.8% required treatment discontinuation; however, significant neutropenia did not occur after 6 weeks of treatment [5]. Ziv et al. reported that 28.8% of children with CCMVI treated with VGCV experienced at least one episode of neutropenia, the majority (84.8%) of which experienced it within the first 3 months of treatment [28]. Moreover, Rawlinson et al. recommended a close neutrophil count follow-up within 6 weeks of treatment initiation in their consensus recommendation [26]. The results of this study are consistent with the abovementioned reports, which have shown that neutropenia is common up to 6 weeks after starting treatment.

As discussed above, neutropenia occurs after VGCV administration, but the reason for individual differences in the timing and degree of neutropenia is unclear. In addition to cytomegalovirus itself [29,30], viral infections such as herpes simplex virus [31], rotavirus [32], enterovirus [33], and measles [34] have previously been reported to cause neutropenia in neonates. However, in our current study, no co-infections with viruses other than CMV were observed. Other agents known to cause neutropenia as an adverse effect include antiepileptic drugs such as phenobarbital, phenytoin, carbamazepine, and valproic acid [35,36,37,38,39]. However, in the NICU setting, the use of these drugs—except for phenobarbital—is limited. Although we investigated the use of phenobarbital in our study population, its overall use was minimal, and in most cases, it was administered only temporarily as a sedative during mechanical ventilation. Moreover, the previous literature suggests that the incidence of phenobarbital-induced neutropenia in neonates is extremely low [40]. Therefore, phenobarbital is unlikely to have contributed significantly to the neutropenia observed in this study. In addition, rare causes of neonatal neutropenia include severe congenital neutropenia due to gene mutations, such as *ELANE*, *HAX1*, and *G6PC3* [41,42,43,44], as well as neonatal alloimmune neutropenia caused primarily by maternally derived anti-neutrophil antibodies [45,46,47,48]. However, in our cohort, neutropenia was transient and resolved after the completion of valganciclovir therapy. Therefore, although no further investigations were performed, we considered the presence of these disorders to be unlikely.

Although few studies on the timing of chemotherapy-induced neutropenia have been conducted, Furuya et al. tested for neutropenia early after treatment initiation (day 8) in different chemotherapy regimens for patients with early-stage breast cancer and reported that neutropenia occurred earlier with docetaxel-based chemotherapy than with other regimens [49], suggesting that the timing of the neutropenic nadir may vary depending on patient characteristics and the drug mechanism of action. Intriguingly, regarding the mechanism by which VGCV causes neutropenia, Billat et al. [50] investigated the association of a selected panel of membrane transporter polymorphisms and the evolution of neutrophil counts in 174 renal transplant recipients and found that a variant of ABCC4 (rs11568658) was associated with decreased neutrophil count following valganciclovir administration. Moreover, HEK293 cells transfected with this ABCC4 variant showed significantly increased intracellular ganciclovir accumulation in in vitro assays, and it was concluded that multidrug resistance-associated protein 4 (MRP4 = ABCC4) controls the intracellular accumulation of ganciclovir and contributes to ganciclovir-induced neutropenia in patients undergoing renal transplant. Therefore, future research that considers the influence of the patient’s genetic background is necessary.

The study findings should be interpreted in light of several limitations. First, although we collected as many cases as possible given the rarity of the disease, the sample size remained small. Therefore, the results should be considered exploratory. Nevertheless, this study specifically targeted severe symptomatic congenital CMV infection and rigorously monitored neutrophil count changes during six months of VGCV treatment, ensuring high data reliability. Second, as a retrospective single-center study, treatment management was not as tightly standardized as in an interventional trial. Consequently, the analysis of factors associated with neutropenia may have been subject to confounding bias. At the same time, the study setting closely reflects real-world clinical practice and should be interpreted as such. In our cohort, different VGCV dosing regimens (16 mg/kg/day vs. 32 mg/kg/day) were used according to the extent of clinical manifestations. Specifically, infants with localized CNS involvement or isolated sensory symptoms (such as ABR or ophthalmologic abnormalities) without systemic signs (e.g., hepatosplenomegaly or petechiae) were treated with 16 mg/kg/day, based on an institutional safety-oriented protocol designed to minimize hematological toxicity. This approach was originally informed by our prior prospective experience [20], which demonstrated acceptable outcomes with this lower dose in a similar patient population. While this dosing strategy may raise concerns about confounding by phenotype, our analysis revealed no significant difference in neutropenia incidence or treatment discontinuation rates between the dosing groups. Furthermore, logistic regression analysis did not identify VGCV dosage as a significant predictor of neutropenia risk. It should also be noted that since the regulatory approval in Japan of oral VGCV at 32 mg/kg/day for 6 months is the standard treatment for symptomatic congenital CMV infection [12], our institution has adopted this dosage universally, regardless of the specific phenotype of disease. Finally, we did not measure drug concentrations over time during VGCV treatment. However, previous reports have shown that there is no correlation between blood drug concentrations and therapeutic outcomes and side effects [14], and that blood concentrations remained stable without major fluctuations owing to the stable pharmacokinetics of oral VGCV [13,51]. In the future, we plan to conduct a prospective study on symptomatic CMV infections treated with VGCV, reporting blood VGCV concentrations and genetic polymorphisms related to drug metabolism as well as neutrophil counts.

## 5. Conclusions

In conclusion, neutropenia occurred in 35% of patients during the 6-week VGCV treatment period, with no marked difference between the 16 mg/kg/day and 32 mg/kg/day dosing groups. Its occurrence was distributed throughout the treatment period and was more frequent in patients with shorter gestational age.

## Figures and Tables

**Figure 1 biomedicines-13-01739-f001:**
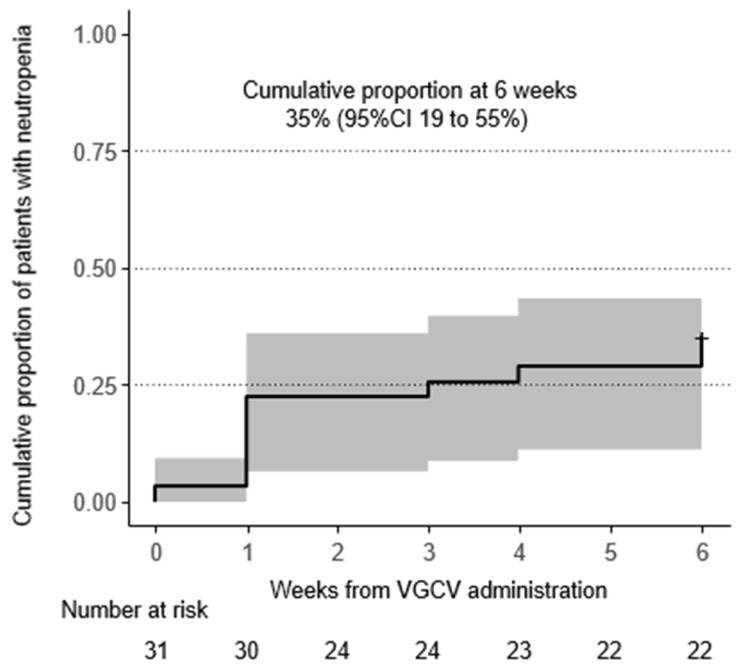
Incidence of neutropenia (neutrophil count < 500/μL): Incidence of neutropenia (neutrophil count < 500/μL) was described using Kaplan–Meier method. The cumulative proportion of patients with neutropenia reached 35% at 6 weeks. Abbreviation: VGCV, valganciclovir; CI, confidence interval.

**Figure 2 biomedicines-13-01739-f002:**
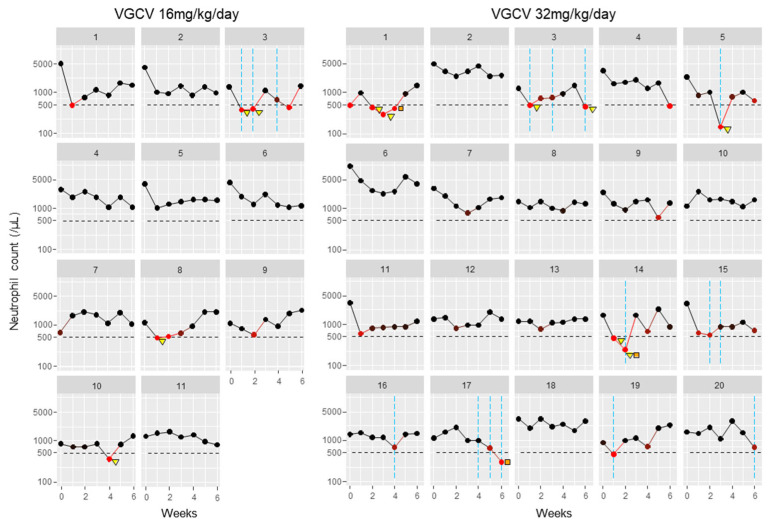
Neutrophil count trajectories in individual patients: Neutrophil counts in individual cases were plotted as circle plots against weeks after treatment initiation, stratified by VGCV dose. Black circles indicate the number of neutrophil counts. Red circles indicate neutrophil counts < 500/mm^3^, with darker red representing counts closer to this threshold. Orange squares indicate VGCV dose reductions, and yellow triangles indicate VGCV discontinuations. Blue vertical lines show the timing of G-CSF treatments. Abbreviation: VGCV, valganciclovir; G-CSF, granulocyte-colony stimulating factor.

**Figure 3 biomedicines-13-01739-f003:**
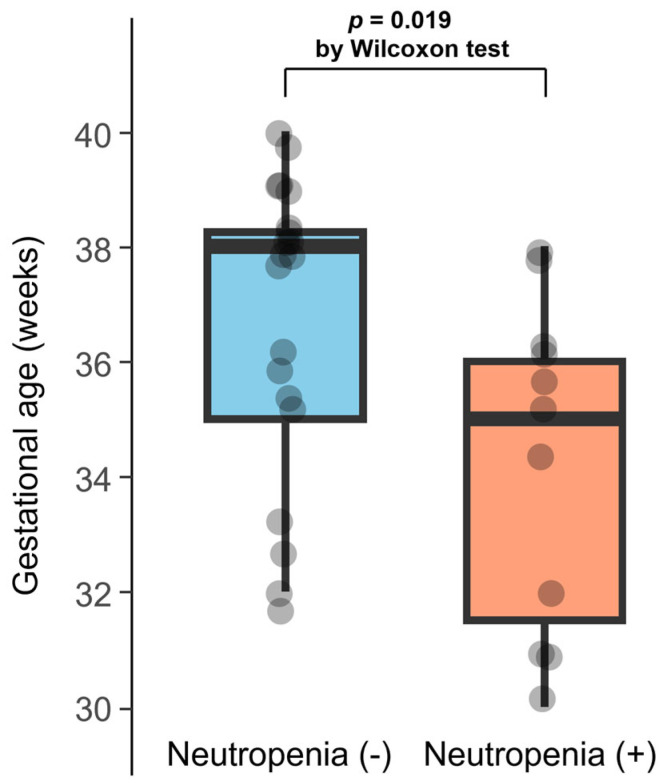
Distribution of gestational age in patients with and without neutropenia at 6 weeks: Gestational age is plotted according to the presence or absence of neutropenia at 6 weeks. The horizontal line within each box indicates the median; the top and bottom edges represent the interquartile range (IQR). Whiskers extend to the most extreme values within 1.5 × IQR from the hinges.

**Table 1 biomedicines-13-01739-t001:** Patient characteristics.

Characteristics	Not Treated with VGCV (*n* = 42)	Total (*n* = 31)	16 mg/kg/day ×6 Weeks (*n* = 11)	32 mg/kg/day ×6 Weeks (*n* = 20)
Gestational age (weeks): median (range)	38 (24 to 41)	36 (30 to 40)	38 (31 to 39)	36 (30 to 40)
Birthweight (g): median (range)	2763 (684 to 3840)	2210 (940 to 3312)	2450 (940 to 3312)	2188 (1255 to 3216)
Male: n (%)	23 (55%)	10 (32%)	4 (36%)	6 (30%)
SGA: n (%)	6 (15%)	11 (35%)	4 (36%)	7 (35%)
Microcephaly: n (%)	3 (8%)	9 (29%)	1 (9%)	8 (40%)
Thrombocytopenia: n (%)	2 (5%)	15 (48%)	2 (18%)	13 (65%)
Liver disfunction: n (%)	2 (5%)	9 (29%)	1 (9%)	8 (40%)
Eye complications: n (%)	0 (0%)	7 (23%)	2 (18%)	5 (25%)
Brain imaging abnormalities: n (%)	6 (14%)	28 (90%)	9 (82%)	19 (95%)
ABR abnormality: n (%)	0 (0%)	22 (71%)	8 (73%)	14 (70%)
Use of phenobarbital	1 (2%)	7 (23%)	0 (0%)	7 (35%)
Initial absolute neutrophil count	2552.5 (560 to 11,640)	1598 (496 to 10,764)	1349 (648 to 5043)	1629 (496 to 10,764)
Blood CMV load before VGCV treatment (copies/mL): median (range)	34.5 (20 to 860)	560 (2.1 to 93,000)	510 (2.4 to 5600)	560 (2.1 to 93,000)
Urine CMV load before VGCV treatment (copies/mL): median (range)	4.0 × 10^6^ (1.1 × 10^5^ to 5.3 × 10^7^)	5.6 × 10^7^ (1.0 × 10^3^ to 1.7 × 10^9^)	4.0 × 10^6^ (1.9 × 10^4^ to 2.1 × 10^8^)	1.1 × 10^8^ (1.0 × 10^3^ to 1.7 × 10^9^)

Abbreviation: SGA, small for gestational age; ABR, auditory brain stem response; CMV, congenital cytomegalovirus; VGCV, valganciclovir.

**Table 2 biomedicines-13-01739-t002:** Summary of events during follow-up.

Characteristics	Total (*n* = 31)	16 mg/kg/day ×6 Weeks (*n* = 11)	32 mg/kg/day ×6 Weeks (*n* = 20)
Neutropenia (neutrophil count < 500/μL): n (%)	11 (35%)	4 (36%)	7 (35%)
Drug interruption: n (%)	7 (23%)	3 (27%)	4 (20%)
Drug reduction: n (%)	3 (10%)	0 (0%)	3 (15%)
Administration of G-CSF: n (%)	9 (29%)	1 (9%)	8 (40%)
Nadir neutrophil count (/µL): median (range)	648 (147 to 2460)	648 (368 to 1058)	647.5 (147 to 2460)
Timing of neutrophil nadir: n (%)			
At week 0	1 (3%)	1 (9%)	0 (0%)
At week 1	6 (19%)	4 (36%)	2 (10%)
At week 2	5 (16%)	1 (9%)	4 (20%)
At week 3	4 (13%)	0 (0%)	4 (20%)
At week 4	4 (13%)	2 (18%)	2 (10%)
At week 5	5 (16%)	1 (9%)	4 (20%)
At week 6	6 (19%)	2 (18%)	4 (20%)

Abbreviation: G-CSF, granulocyte-colony stimulating factor.

**Table 3 biomedicines-13-01739-t003:** Exploration of patient factors associated with neutropenia using univariable logistic regression analyses.

Characteristics	Odds Ratio for Neutropenia	95% Confidence Interval	*p*-Value
VGCV 32 mg/kg (ref. 16 mg/kg)	0.94	0.20 to 4.37	0.939
Gestational age (1 week increase)	0.72	0.54 to 0.97	0.031
Birthweight (100 g increase)	0.95	0.84 to 1.08	0.451
Male (ref. female)	1.33	0.28 to 6.33	0.717
SGA (ref. absence)	0.56	0.11 to 2.79	0.481
Microcephaly (ref. absence)	0.88	0.17 to 4.49	0.873
Thrombocytopenia (ref. absence)	1.47	0.33 to 6.43	0.612
Liver disfunction (ref. absence)	0.41	0.07 to 2.46	0.332
Eye complications (ref. absence)	0.67	0.11 to 4.18	0.665
Brain imaging abnormalities (ref. absence)	1.11	0.09 to 13.84	0.935
ABR abnormality (ref. absence)	1.14	0.22 to 5.87	0.873
Blood CMV load before VGCV treatment (10 times increase)	1.3	0.61 to 2.79	0.499
Urine CMV load before VGCV treatment (10 times increase)	1.04	0.61 to 1.76	0.891

Abbreviations: SGA, small for gestational age; ABR, auditory brain stem response; CMV, congenital cytomegalovirus; VGCV, valganciclovir.

## Data Availability

The original contributions presented in this study are included in the article/Appendix A. Further inquiries can be directed to the corresponding author.

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
