# Peer review of "Changes in Neutrophil Count During Valganciclovir Therapy for Symptomatic Congenital Cytomegalovirus Infection"

_biomedicines, 2025, doi:10.3390/biomedicines13071739_

Round 1

Reviewer 1 Report

Comments and Suggestions for Authors

The authors present a single center retrospective review of infants seen over an 8 year period 2009-2017, diagnosed with virologically confirmed, symptomatic congenital CMV infection (CCMVI) who were treated with valganciclovir (VGCV), various treatment protocols, with a focus on occurrence of neutropenia within the first 6 weeks of treatment. They concluded neutropenia occurred 35% of their population within 6 weeks of treatment, with a variety of patterns, and younger gestation age may be associated with neutropenia. 

1) Title- adequately describes paper contents

2) Good representation of findings in paper

3) Introduction-  informative

4) Materials and Methods- line 69 " Infants aged > 60 days at the start of VGCV " infection " " maybe meant to say " treatment " - suspect there is a typo here ; do the authors have access to a control or comparison group of infants with CCMVI who were not treated with VGCV? It may be insightful, especially since CCMVI can also be associated with neutropenia in some infants, as well as other things can cause neutropenia, such as intercurrent viral illnesses, other medications such as seizure medications, development of anti neutrophil antibodies, underlying neutropenia syndromes, etc. ; were concurrent medications also tracked and analyzed or available for analysis? were intercurrent viral illnesses tracked or monitored or captured in the data analyzed? did any patients get tested for anti neutrophil antibodies? The authors analyzed the degree of CMV viremia levels in blood plasma as a correlation of neutropenia occurrence in Table 3, but did they follow CMV viremia levels on treatment and find any correlation with persistence of CMV viremia and neutropenia? can the authors provide rationale why more severe CCMVI with CNS and sensory involvement received lower reduce dosage regimen of VGCV, stated in lines 104/105? And while no differences were found between the two dosing regimens,  is it still possible two different dosing regimens for two different phenotypes confounded the results?

5) Results-  line 152/153 the authors state time to first onset of neutropenia, occurrences were most frequent on the first day of VGCV treatment- if this is so, how can the authors attribute VGCV as the cause/contributor? Is it more likely the CCMVI infection itself or another factor ( medications, other viruses, etc) were contributing to the neutropenia present at onset of VGCV treatment? Also, review of Figure 2 shows only 2 patients with neutropenia 500 or below on the graph at the start of treatment, so does not appear to be the most common . Can the authors reconcile this apparent discrepancy in data presented in Figure 2 and the statements made in the text on lines 152/153? Neutropenia that occurs after 1-2 weeks of VGCV is more likely attributable to the antiviral itself than neutropenia that is present at the start first day of VGCV treatment. The authors should comment on this possibility. Can the authors analyze neutropenia that occurred after 1-2 weeks GVCG treatment in the same manner as overall neutropenia? It appears in Figure 1 the Kaplan Meier curve, that neutropenia cumulative is significant at starting 1-2 weeks into treatment. Did any of the patients experience invasive infections or other symptoms as a consequence of their neutropenia? Or was the neutropenia a laboratory observed event only? 

6) Discussion-covers the main point(s) of the paper well; it may benefit from some re focusing and shortening; for example lines 221-242 the authors , rightly so, state the reason for individual differences in the timing and degree of neutropenia appear unclear from their study, they should mention other concomitant factors, such as medications, viruses, immune and auto conditions; also while interesting, the in depth discussion 222-242 on chemotherapy associated neutrophil kinetics and genetics may be a bit far afield of the relevant contents for this paper; perhaps the authors can briefly mention with a sentence or two w/ a reference the other possibilities of genetic causes and suggest personal genome studies may reveal some infants at higher risk for neutropenia than others, and may be avenues for future research. But before they assume genetics, they may wish to also make sure other causes, mentioned above, are analyzed. 

7) Tables and Figures - appear appropriate and for most part easy to understand and add to the manuscript

Author Response

Comments 1: The authors present a single center retrospective review of infants seen over an 8 year period 2009-2017, diagnosed with virologically confirmed, symptomatic congenital CMV infection (CCMVI) who were treated with valganciclovir (VGCV), various treatment protocols, with a focus on occurrence of neutropenia within the first 6 weeks of treatment. They concluded neutropenia occurred 35% of their population within 6 weeks of treatment, with a variety of patterns, and younger gestation age may be associated with neutropenia. 

1) Title- adequately describes paper contents

2) Good representation of findings in paper

3) Introduction-  informative

Response 1:

Thank you very much for your positive and encouraging comments regarding our manuscript. We are grateful for your recognition of the title, findings, and introduction, and appreciate your support.

Comments 2:

4) Materials and Methods

- line 69 " Infants aged > 60 days at the start of VGCV " infection " " maybe meant to say " treatment " - suspect there is a typo here ;

Response 2:

Thank you for pointing out the typographical error. We have corrected the wording from "infection" to "treatment" in line 69 to accurately reflect the intended meaning.

Comments 3:

Do the authors have access to a control or comparison group of infants with CCMVI who were not treated with VGCV? It may be insightful, especially since CCMVI can also be associated with neutropenia in some infants, as well as other things can cause neutropenia, such as intercurrent viral illnesses, other medications such as seizure medications, development of anti neutrophil antibodies, underlying neutropenia syndromes, etc. ;

were concurrent medications also tracked and analyzed or available for analysis?

were intercurrent viral illnesses tracked or monitored or captured in the data analyzed? did any patients get tested for anti neutrophil antibodies?

Response 3:
Thank you for your insightful suggestion. In response, we included a comparison group of infants with asymptomatic CCMVI who did not receive VGCV treatment. In this untreated control group, there were fewer complications and a tendency toward lower neutrophil counts and viral loads compared to the treatment group. However, since the timing of sample collection was not consistent between the two groups, a direct comparison could not be performed. Nevertheless, including this group allowed us to better assess the contribution of VGCV to neutropenia, independent of CMV infection itself.

With regard to potential confounding factors:

  • We reviewed the medical records and found no evidence of intercurrent viral infections in any of the patients during the observation period.
  • We also reanalyzed the frequency of use of phenobarbital, a sedative and anticonvulsant commonly administered in NICUs, which has been reported to potentially cause neutropenia. However, previous literature suggests that the incidence of phenobarbital-induced neutropenia in neonates is extremely low (Brain Dev. 1993 Jul-Aug;15(4):258–262. doi: 10.1016/0387-7604(93)90020-9), and we therefore consider it unlikely to be a major contributor in our cohort.
  • None of the patients developed persistent or prolonged neutropenia after the completion of VGCV treatment. Therefore, tests for anti-neutrophil antibodies and evaluations for congenital neutropenia syndromes were not actively performed, as no clinical indications arose during follow-up.

We have incorporated these clarifications into the revised manuscript.

Comments 4:

The authors analyzed the degree of CMV viremia levels in blood plasma as a correlation of neutropenia occurrence in Table 3, but did they follow CMV viremia levels on treatment and find any correlation with persistence of CMV viremia and neutropenia?

Response 4:

As shown in Table 3, we did not find a significant correlation between CMV viral load at baseline and the occurrence of neutropenia. Additionally, we conducted further analyses to assess the relationship between viral load and neutropenia occurring at two weeks after treatment initiation. These analyses similarly did not demonstrate any association between viral load and the development of neutropenia (Supplemental Data). Furthermore, in our previous study, we reported that changes in viral load during treatment did not correlate with auditory outcomes (Kido et al., JCM 2021).

Comments 5:

Can the authors provide rationale why more severe CCMVI with CNS and sensory involvement received lower reduce dosage regimen of VGCV, stated in lines 104/105? And while no differences were found between the two dosing regimens, is it still possible two different dosing regimens for two different phenotypes confounded the results?

Response 5:

Thank you for your important and thoughtful comment. We agree that the assignment of different VGCV dosing regimens could potentially introduce confounding by indication. In our study, the rationale for using a lower dose regimen (16 mg/kg/day) in patients with central nervous system (CNS) or sensory organ involvement was based on our institutional protocol, which aimed to reduce the risk of hematological toxicity, especially neutropenia, in infants without generalized systemic symptoms (e.g., hepatosplenomegaly or petechiae). This strategy was informed by our previous prospective study (Nishida et al., Brain Dev, 2016), in which infants with limited CNS or auditory/ophthalmological involvement were successfully treated with a 16 mg/kg/day regimen, while closely monitoring viral load and clinical response. This approach prioritized safety in patients who were often stable but still met criteria for symptomatic CCMVI.

Importantly, in the current study, we observed no significant difference in the incidence of neutropenia or treatment discontinuation between the two dosing groups. Furthermore, the dosing group (32 mg/kg/day vs. 16 mg/kg/day) was not identified as a significant predictor of neutropenia in the univariable logistic regression analysis. These findings suggest that the use of different dosing regimens for different clinical phenotypes did not substantially confound our analysis of neutropenia risk. Nonetheless, we acknowledge that this remains a limitation of our observational study, and we have added a note about this point in the discussion section to address potential confounding by indication.

Additionally, we would like to note that since the formal approval of oral VGCV (32 mg/kg/day for 6 months) for symptomatic congenital CMV infection in Japan, we have adopted this dosage as the standard of care in all cases, regardless of symptom phenotype.

Comments 6:

5) Results- line 152/153 the authors state time to first onset of neutropenia, occurrences were most frequent on the first day of VGCV treatment- if this is so, how can the authors attribute VGCV as the cause/contributor?

Response 6:

Thank you for pointing this out. We apologize for the error — the correct description should be “the first week of VGCV treatment,” not “the first day.” We have revised the manuscript accordingly.
We believe that the fact that neutropenia frequently developed shortly after the initiation of VGCV supports the possibility that VGCV contributed to its onset.

Comments 7:

Is it more likely the CCMVI infection itself or another factor (medications, other viruses, etc) were contributing to the neutropenia present at onset of VGCV treatment?

Response 7:

Thank you for your important comment. As we noted in our response to the previous question, multiple factors could potentially contribute to neutropenia in this patient population. We will address this point more explicitly in the Discussion section, including the possible roles of congenital CMV infection itself, concomitant medications, and other viral infections.

Comments 8:

Also, review of Figure 2 shows only 2 patients with neutropenia 500 or below on the graph at the start of treatment, so does not appear to be the most common. Can the authors reconcile this apparent discrepancy in data presented in Figure 2 and the statements made in the text on lines 152/153? Neutropenia that occurs after 1-2 weeks of VGCV is more likely attributable to the antiviral itself than neutropenia that is present at the start first day of VGCV treatment. The authors should comment on this possibility.

Response 8:

Thank you for your insightful comment. As you correctly pointed out—and as noted in our previous response—the original text mistakenly referred to the “first day” of treatment, when in fact, neutropenia most frequently occurred during the first week after VGCV initiation. We have corrected this statement in the revised manuscript.

Furthermore, we fully agree with your observation that neutropenia occurring after 1–2 weeks of treatment is more likely attributable to the effects of VGCV itself, rather than pre-existing conditions. We have revised the relevant sections of the manuscript accordingly to clarify this point and avoid confusion.

Comments 9:

Can the authors analyze neutropenia that occurred after 1-2 weeks VGCV treatment in the same manner as overall neutropenia? It appears in Figure 1 the Kaplan Meier curve, that neutropenia cumulative is significant at starting 1-2 weeks into treatment.

Response 9:

As mentioned earlier, we conducted a separate analysis of neutropenia occurring within the first 1–2 weeks after the initiation of VGCV treatment, using the same analytical methods as for overall neutropenia. This analysis yielded the same result, identifying gestational age as the only significant risk factor.

Comments 10:

Did any of the patients experience invasive infections or other symptoms as a consequence of their neutropenia? Or was the neutropenia a laboratory observed event only? 

Response 10:

Thank you for your important question. None of the patients in our cohort developed severe infections attributable to neutropenia during the treatment course. We believe that one contributing factor may be that all patients received the 6-week VGCV treatment while hospitalized in either the NICU or GCU, which likely minimized exposure to community-acquired pathogens. Therefore, neutropenia in this study was a laboratory-observed event without apparent clinical consequences.

Comments 11:

6) Discussion-covers the main point(s) of the paper well; it may benefit from some re focusing and shortening; for example lines 221-242 the authors , rightly so, state the reason for individual differences in the timing and degree of neutropenia appear unclear from their study, they should mention other concomitant factors, such as medications, viruses, immune and auto conditions; also while interesting, the in depth discussion 222-242 on chemotherapy associated neutrophil kinetics and genetics may be a bit far afield of the relevant contents for this paper; perhaps the authors can briefly mention with a sentence or two w/ a reference the other possibilities of genetic causes and suggest personal genome studies may reveal some infants at higher risk for neutropenia than others, and may be avenues for future research. But before they assume genetics, they may wish to also make sure other causes, mentioned above, are analyzed. 

Response 11:
Thank you for your valuable suggestions. In accordance with your comment, we have deleted the discussion on chemotherapy-associated neutropenia. We also added a paragraph discussing other potential causes of neutropenia, such as concomitant medications, viral infections, and immune-related mechanisms.

Comments 12:

7) Tables and Figures - appear appropriate and for most part easy to understand and add to the manuscript

Response 12:
Thank you very much for your positive evaluation.

Reviewer 2 Report

Comments and Suggestions for Authors

The manuscript titled “Changes in neutrophil count during valganciclovir therapy for symptomatic congenital cytomegalovirus infection” described the incidence of valganciclovir induced neutropenia and associated factors. The study found that shorter gestational age was associated with this outcome. I have the following comments,

  1. The study content was very interesting and helpful for pediatricians who may have to use this drug.
  2. The following factors should also be taken into account,

2.1. the duration of neutropenia

2.2. the initial absolute neutrophil count

2.3. any consequences of neutropenia

  1. In the result part, the authors stated that “Regarding the time to first onset of neutropenia, occurrences were most frequent on the first day of valganciclovir treatment,” Is it actually the first week instead of the first day?
  2. The figure 2 was quite difficult to read especially orange squares and yellow triangles. They could be hardly seen.
  3. Why were more proportions of patients having drug reduction and administration of G-CSF in the group of 32 mg/kg/day?
  4. In the discussion part, the contents discussing mechanisms of other chemotherapeutic drugs induced neutropenia were quite irrelevant to the context of the study. Please correct accordingly.

Author Response

The manuscript titled “Changes in neutrophil count during valganciclovir therapy for symptomatic congenital cytomegalovirus infection” described the incidence of valganciclovir induced neutropenia and associated factors. The study found that shorter gestational age was associated with this outcome. I have the following comments,

Comments 1:

  1. The study content was very interesting and helpful for pediatricians who may have to use this drug.

Response 1:

Thank you very much for your positive and encouraging comment. We are pleased to know that our study could be of help to pediatricians in clinical practice.

Comments 2:

  1. The following factors should also be taken into account,

2.1. the duration of neutropenia

2.2. the initial absolute neutrophil count

2.3. any consequences of neutropenia

Response 2:
Thank you for your insightful comments and valuable suggestions.

As shown in Figure 2, the duration of neutropenia generally corresponded with the VGCV treatment period. In all cases, neutropenia was transient and resolved promptly following appropriate clinical interventions, such as dose reduction or temporary discontinuation of VGCV. No adverse events or clinical complications directly attributable to neutropenia were observed.

Regarding the initial absolute neutrophil count (ANC), we reanalyzed the data, including comparisons between the treatment group and an untreated, non-CCMVI control group. In the untreated control group, there was a trend toward lower initial ANC and fewer complications compared to the treatment group. However, due to differences in the timing of sample collection between groups, a direct comparison was not feasible.

Comments 3:

  1. In the result part, the authors stated that “Regarding the time to first onset of neutropenia, occurrences were most frequent on the first day of valganciclovir treatment,” Is it actually the first week instead of the first day?

Response 3:

Thank you very much for pointing this out. You are correct—the correct expression should be "the first week" rather than "the first day." We have revised the manuscript accordingly to reflect this correction.

Comments 4:

  1. The figure 2 was quite difficult to read especially orange squares and yellow triangles. They could be hardly seen.

Response 4:
Thank you for your valuable feedback regarding Figure 2. We apologize for the difficulty in readability, particularly with the orange squares and yellow triangles. We have revised the figure to improve clarity and visibility of all data points. Please review the updated version.

Comments 5:

  1. Why were more proportions of patients having drug reduction and administration of G-CSF in the group of 32 mg/kg/day?

Response 5:

Thank you for your important comment.
Since we used two different dosing regimens—16 mg/kg/day and 32 mg/kg/day—the management strategy for neutropenia varied between the two groups. In the 32 mg/kg/day group, dose reduction or G-CSF administration was feasible options when neutropenia occurred. In contrast, for patients receiving the lower dose of 16 mg/kg/day, further dose reduction was not considered practical, and drug interruption was often the only available option.
However, the overall rate of treatment discontinuation due to neutropenia did not differ significantly between the two groups. Moreover, in the univariable logistic regression analysis, the 32 mg/kg/day regimen was not associated with an increased risk of neutropenia. Therefore, we believe that this difference in G-CSF use or dose adjustment does not represent a fundamental issue.

Comments 6:

  1. In the discussion part, the contents discussing mechanisms of other chemotherapeutic drugs induced neutropenia were quite irrelevant to the context of the study. Please correct accordingly.

Response 6:

Thank you for your valuable comment. In accordance with your suggestion, we have removed most of the discussion related to the mechanisms of neutropenia induced by other chemotherapeutic agents, as we agree that it was not directly relevant to the context of our study.

Round 2

Reviewer 2 Report

Comments and Suggestions for Authors

The issues have been well addressed.